# Embedded Graphite and Carbon Nanofibers in a Polyurethane Matrix Used as Anodes in Microbial Fuel Cells for Wastewater Treatment

**DOI:** 10.3390/polym15204177

**Published:** 2023-10-21

**Authors:** Pedro Pérez-Rodríguez, Carlos A. Covarrubias-Gordillo, José A. Rodríguez-De la Garza, Cynthia L. Barrera-Martínez, Silvia Y. Martínez-Amador

**Affiliations:** 1Departamento de Ciencias del Suelo, Universidad Autónoma Agraria Antonio Narro, Calzada Antonio Narro 1923, Buenavista, Saltillo 25315, Coahuila, Mexico; pedro.perezr@uaaan.edu.mx; 2Departamento de Materiales Avanzados, Centro de Investigación en Química Aplicada, Boulevard Enrique Reyna Hermosillo 140, San José de los Cerritos, Saltillo 25113, Coahuila, Mexico; carlos.covarrubias@ciqa.edu.mx; 3Departamento de Biotecnología, Facultad de Ciencias Químicas, Universidad Autónoma de Coahuila, José Cárdenas Valdez y Venustiano Carranza S/N, Colonia República Oriente, Saltillo 25280, Coahuila, Mexico; antonio.rodriguez@uadec.edu.mx; 4Centro de Investigación para la Conservación de la Biodiversidad y Ecología de Coahuila, Universidad Autónoma de Coahuila, Miguel Hidalgo 212, Zona Centro, Cuatrociénegas 27640, Coahuila, Mexico; cynthia_barrera@uadec.edu.mx; 5Departamento de Botánica, Universidad Autónoma Agraria Antonio Narro, Calzada Antonio Narro 1923, Buenavista, Saltillo 25315, Coahuila, Mexico

**Keywords:** polyurethane, graphite, carbon nanofibers, microbial fuel cell, bioelectrochemical system, municipal wastewater

## Abstract

Composites of polyurethane and graphite and polyurethane and carbon nanofibers (PU/Graphite 0.5% and PU/CNF 1%) were synthesized and used as anodes in dual-compartment microbial fuel cells (MFCs) for municipal wastewater treatment; electrical energy generation and organic matter removal were assessed. The maximum power density, coulombic efficiency and chemical oxygen demand (COD) removal efficiency in the MFCs packed with the PU/Graphite 0.5% and PU/CNF 1% composites were 232.32 mW/m^3^ and 90.78 mW/m^3^, 5.87 and 4.41%, and 51.38 and 68.62%, respectively. In addition, the internal resistance of the MFCs with the best bioelectrochemical performance (PU/Graphite 0.5%) was 1051.11 Ω. The results obtained in this study demonstrate the feasibility of using these types of materials in dual-compartment MFCs for wastewater treatment with electric power generation.

## 1. Introduction

Microbial fuel cells (MFC) are bioelectrochemical devices (bioelectrochemical systems) capable of converting chemical energy, contained in a wide variety of substrates, to electrical energy through electrochemically active microorganisms (commonly called exoelectrogens). These microorganisms can transfer the energy generated in the system to an external support for its use [1]. This technology is of great interest when considering its ability to generate electricity by degrading contaminated effluents [2,3].

The performance of bioelectrochemical systems (BESs) depends on multiple factors, such as the conductivity and microbial biocompatibility of the selected electrodes [4], the nature and concentration of the substrate [5], the absence or presence of a cation exchange membrane [6] and the overall design of the system [7].

Considering that the electrodes (anode and cathode) are where the transfer of electrons occurs and that the microorganisms that inhabit the system grow and reproduce on their surface, several authors conclude that their correct selection defines, in greater proportion, the system’s performance [8]. For this reason, various materials have been studied to manufacture electrodes with greater conductivity and microbial affinity, among which metals, carbonaceous materials and conductive polymers stand out [9,10,11]. Much of the current research has been directed at developing and characterizing electrodes based on innovative composites to improve the bioelectrochemical capabilities of MFC technology. For example, Wang et al. [12] developed anodes from metal–organic structures of iron oxide on a carbon felt matrix. These materials have been shown to increase the electrochemical capacities and the biocompatibility of the anodes in the MFCs, promoting the adhesion of electrically active microorganisms and favoring the secretion of proteins related to the extracellular transfer of electrons. Liu et al. [13] synthesized nitrogen-doped carbon nanofiber anodes for anchoring iron nanoparticles, evaluating electrical power generation in microbial fuel cells. When carrying out the morphological and structural characterization of the materials, they found porous structures uniformly distributed in the carbon nanofiber matrix, which favored microbial colonization in the internal part of the support. Yang et al. [14] evaluated organic matter and ammonia removal in sediment microbial fuel cells packed with graphene oxide-coated carbon cloth composites as anodes. The results showed that by applying these materials, the presence of exoelectrogenic microorganisms of the genera *Sulfurovum* and *Lactobacillus* was enriched, thus increasing the electrochemical performance and the removal of contaminants in the system. Zhu et al. [15] applied a coating of carbon nanoparticles doped with N, P, S and Co heteroatoms to a carbon cloth matrix as anodes in microbial fuel cells. The authors selected these materials due to their high biocompatibility, high conductivity, low cost and functionalization. It was observed that the increase in the surface area and the porosity of the support promoted the formation of microbial biofilm, the consumption of the substrate and the generation of electrical energy. Moradian et al. [16] developed a carbon felt anode superficially modified with polyaniline nanofibers, intending to increase bioelectricity and hydrogen production using xylose as a substrate and using *Cystobasidium slooffiae* JSUX strain as the inoculum in a microbial fuel cell. The results suggested that the extracellular electron transfer process and energy production are maximized when using this type of composite material in microbial fuel cells. Yaqoob et al. [17] built composites (derived from cellulosic waste) of polyaniline-coated graphene oxide (GO-PANI) to promote the electron transfer rate in benthic microbial fuel cells (BMFCs), observing a four-fold performance in the modified supports (GO-PANI) when compared to the uncoated material. Nishio et al. [18] used urethane composites coated with activated carbon and carbon nanotubes as anodes in microbial fuel cells. The authors detected a considerable increase in the material’s conductivity after the coating was carried out, confirming the capability of this type of support in bioelectrochemical systems to generate electrical energy.

Therefore, it can be seen that a correct selection of the anodic material will promote the generation of electrical energy and the degradation of the substrate in microbial fuel cells. Previous work has been focused on the development of innovative coatings that increase the conductivity and biocompatibility of the modified materials, but due to their low mechanical resistance and poor adhesion to the base material, this research has been oriented towards the embedding of the materials in a highly porous matrix. The present study focused on synthesizing graphite and carbon nanofiber-based composites embedded in a polyurethane matrix. The composites were used as anodes in microbial fuel cells during municipal wastewater treatment. The generation of electrical energy in the system and the removal of organic matter contained in municipal wastewater were evaluated. This work provides a convenient and practical method to fabricate anodes with high surface area and low electrical resistivity, which can be used to increase the performance of bioelectrochemical systems.

## 2. Materials and Methods

### 2.1. Materials

The graphite was purchased from Sigma-Aldrich (Toluca, Mexico), and it had a diameter greater than 20 microns and a purity of 99%. The carbon nanofibers (CNFs) were supplied by Pyrograf Products, Inc. (Cedarville, OH, USA); these were thermally treated at 3000 °C to increase the degree of graphitization (PR-24-XT-HHT). In addition, they have an average diameter of 100 nm, lengths from 50 to 200 microns, a surface area of 41 m^2^/g and a purity greater than 95%. The mixture of urethanes and 4-4 diphenylmethane disocyanate to obtain the polyurethane foam was purchased from Especialidades Químicas para el Poliéster, S.A. de C.V. (Nezahualcóyotl, Mexico State, Mexico). Sulfuric acid (96~98%), potassium dichromate (96~98%), mercury sulfate (>98%), silver sulfate (>98%) and potassium biphthalate (>99.95%), used during the determination of chemical oxygen demand (COD), were purchased from FERMONT (Monterrey, Mexico). The cation exchange membrane used in the MFCs was purchased from Membranes International Inc. (Ringwood, NJ, USA) (CXM-200, standard thickness 0.45 ± 0.025 mm).

### 2.2. Preparation of PU/Graphite and PU/CNF Composites

For the manufacture of foamed polymers composed of 0.5% wt./wt. of graphite and 1% wt./wt. of CNFs (concentrations selected from the maximum loading percentage of the conductive materials in the polyurethane matrix), 1.3 g of graphite and 2.6 g of carbon nanofibers were dispersed in 150 mL of polyol by mechanical stirring until homogeneous. Subsequently, 64 mL of disocyanate was added to each sample and mixed for 60 s with a double-blade homogenizer at 4000 rpm. The already foamed system was allowed to settle for 25 min. Finally, the foam was cured in an oven at 100 °C for 4 h (Figure 1). Table 1 shows the composition used to prepare each composite.

### 2.3. Material Characterization

Figure 1 shows a diagram of the synthesis process and the characterization of the manufactured materials. For the qualitative identification of the components of the 0.5% PU/Graphite and 1% PU/CNF anodes, infrared spectra of the samples of each of the components were carried out. A Nicolet Magna 550 spectrophotometer was used in a wavelength range of 4000 cm^−1^ to 400 cm^−1^. Previously, the samples were dried in a vacuum oven at 100 °C for 15 h. The spectra obtained were normalized in absorbance and subsequently analyzed in transmittance to discuss the results. To analyze the morphology and the state of dispersion and spatial distribution of the particles (graphite and CNF) in the PU/Graphite 0.5% and PU/CNF 1% anodes, a SEM analysis was performed using a scanning electron microscope JEOL JMS-7401F (Akishima, Tokyo, Japan) field emission detector, applying a voltage of 6 kV using a secondary electron detector at a working distance between the objective lens and the sample of 6.0 mm. The samples analyzed were prepared from cryogenic fractures of the composites. The cross-section of the fracture fragments was coated with Au-Pd for further analysis. Finally, for the identification of the electrical conductivity of the samples, an electrical resistivity analysis was carried out in a desktop digital teraohmmeter of the Keithley brand, model 6517B (Tektronix, Beaverton, OR, USA). The tests were carried out at 500 V, in a range of 2 µA.

### 2.4. MFC Construction and Operation

Figure 2 shows the configuration of the MFCs used in the experiments. Double-compartment microbial fuel cells were used, with an approximate volume of 2000 mL (1000 mL in each compartment), adding a piece of graphite felt (10 × 10 × 0.8 cm) in the cathode compartment (cathode) and 1000 mL of deionized water as the cathodic solution; this compartment was externally oxygenated with a Boyu brand air pump (11.6 cm long × 7 cm wide × 5.6 cm high) coupled to an Imagitarium brand 4-way air control valve (7.6 cm long × 5.8 cm wide × 10.9 cm high). One of the previously synthesized supports (anode) (PU, PU/Graphite 0.5% and PU/CNF 1%) was placed in the anode compartment, with 1000 mL of raw municipal wastewater (without prior treatment) as substrate and inoculum (the physicochemical characteristics of wastewater are shown in Table 2). A pre-hydrated cation exchange membrane (in 5% NaCl solution for 12 h at 40 °C) was used to separate the compartments. The distance between the anode and cathode was 3 cm.

Cell monitoring was carried out using a digital multimeter (Fluke 289—Trendcapture) with which the voltage (V) of each cell was determined 2 times a day (morning and night) during the entire reaction (48 days, from which the first 30 days were for the development of the biofilm on the surface of the anodes, and on day 31, the MFCs were fed with recently collected municipal wastewater and the monitoring of the kinetics began), using an external resistance of 1 kΩ to close the circuit and a stainless steel wire as an electron collector.

The volumetric power density (mW/m^3^) generated in the MFCs was calculated as
(1)$$P_{V}=\frac{UI}{V}*1000$$
where *U* is the voltage (V), *I* is the electric current (A) and *V* is the volume of the anode compartment [19]. On the other hand, the coulombic efficiency (%) was calculated as
(2)$$E_{Cb}=\frac{M\int_{0}^{t}Idt}{FbV_{An}\Delta COD}$$
where *M* = 32 is the molecular weight of oxygen, *I* is the electric current (calculated from the voltage generated by the MFC), *F* = 96,485.33 C/mol is Faraday’s constant, *b* = 4 is the number of electrons exchanged per mole of oxygen, *V_An_* is the volume of the substrate in the anode compartment (1 L) and Δ*COD* is the difference in COD over time [20]. Finally, the chemical oxygen demand (COD) was determined at the beginning and end of the reaction to evaluate the removal of organic matter in the system [21].

### 2.5. MFC Electrochemical Characterization

The variable resistance method was used to determine the polarization and power density curves, which consisted of applying an external resistance to the system that varied from 50 × 10^−3^ to 100 kΩ once the voltage stabilized after feeding the system with a new sample of raw municipal wastewater and waiting for the open-circuit voltage (OCV) to reach a steady value. The power density obtained by this method was normalized to the volume of the anode compartment in m^3^.

## 3. Results and Discussion

### 3.1. Materials and Composites Analysis

#### 3.1.1. Infrared Chemical Composition Analysis

Figure 3 shows the infrared spectrum of the fabricated materials. The main bands found are typical of a PU matrix reinforced with carbon-based particles; the presence of carboxylic groups on the strong bands is observed at 1719 and 1089 cm^−1^. The two bands observed between 2976 and 2863 cm^−1^ were attributed to symmetric and non-symmetric stretching of the CH_2_ bond. Polymerized urethanes are in the regions of 3334 and 1225 cm^−1^ and 1502–1543 cm^−1^. These bands are typical of the stretching C=O and NH bonds [22,23,24]. Finally, it is worth mentioning that there was no hydroxyl peak at 3450 and 2312 cm^−1^, indicating that all isocyanates contained in the pre-polymer constituents were fully used, which translates into the null existence of OH groups in the composites. An excess of OH groups favors crosslinking and a more flexible polymeric structure with a variable degree of branching and, consequently, different physical properties [25,26]. The above suggests that CNFs and graphite do not intervene in the foaming reaction, altering the ratio of isocyanate and polyol in the formulation. However, they can influence the mechanical properties by intervening in the nucleation and, consequently, the morphology of the final composite [27].

#### 3.1.2. Scanning Electron Microscopy Morphological Analysis

Figure 4 shows the morphology of the PU foam matrix synthesized from the polyol with isocyanate mixture. In general, it can be observed that the foams have an open cell structure, which increases the surface area and the transfer of filtered matter through the material. However, domains of the polymeric matrix can be observed without completing the structure of the cell in its entirety. On the other hand, the system’s porosity is not very homogeneous and has pores ranging from 0.04 mm to 0.5 mm. 

Figure 5 shows the composites made with 0.5% wt./wt. of graphite in different magnifications. In Figure 5a, it can be seen that graphite (at this concentration) preserves the cellular structure of PU to a greater extent, so the material has lower density and resistance to flow. However, short domains of the polymeric matrix are still observed because most of the pores tend to end in a break in the cellular structure. In Figure 5b, it can be seen (circled in red) that the added particles could function as points of pore generation or thinning of the polymeric matrix, while Figure 5c shows that the added graphite is anchored to the surface of the PU matrix and that this generates roughness and microcracks in the polymeric matrix, which could lead to the porosity of the cellular structure.

Figure 6 shows the composites made with 1% wt./wt. of CNFs at different magnifications. It is observed (Figure 6a) that the use of nanoparticles shortens the domains of the polymer matrix, preventing the formation of a cellular structure in the system, decreasing the volume of the material obtained and increasing its density and resistance to the flow of a medium through the material. In addition, it is observed that the polymeric domains have a higher roughness in relation to the pure PU matrix. Such roughness is suggested to be formed by the dispersion of the agglomerates and nanoparticles of the CNFs on the surface of the polymeric matrix during the foaming reaction (see Figure 6b) [28]. In Figure 6c, it can be observed that the forces during foaming that occur in the composite formation reaction generate the homogeneous dispersion of the nanoparticles; however, it causes the rupture of the CNF agglomerates to different degrees, causing the CNFs to be obtained individually (enclosed in red circles), as well as agglomerates of various sizes (framed in red dotted boxes). Quiang et al. [29] found that adding carbon particles produces roughness on the surface of the polymer matrix, which results in a greater surface area that could be used in absorption applications. Similar results were obtained by Shi et al. [28], mentioning that carbon-based particles could be firmly anchored to the surface of a PU matrix, making it rougher. In addition, they mention that a smaller particle has a higher velocity (higher kinetic energy) in the dispersion promoted by the foaming process, thus leading to a greater adhesion of the particles to the PU foam. Additionally, various reports have mentioned that PU foam microstructures in which carbon particles are added trigger heterogeneous cell nucleation and decrease cell size [30,31], suggesting that the observed structural differences are mainly due to the dispersion obtained depending on the carbon structure used.

#### 3.1.3. Electrical Conductivity Analysis

The electrical conductivity obtained by the composites of PU/Graphite 0.5% and PU/CNF 1% was 1.54 × 10^−9^ S/m and 1.13 × 10^−9^ S/m, respectively. When conductive particles/nanoparticles (such as carbon-based ones) are added to a polymer matrix after a specific concentration of particles is reached, an interconnecting network is generated and promotes the electron flow through the material, known as tunneling. However, this effect only succeeds if the particles/nanoparticles are in contact or have relatively short distances between them. For this specific case, a slight increase in electrical conductivity can be attributed to the fact that the CNFs and graphite have a high dispersion and distribution, hindering the formation of conductive pathways. Also, it is worth mentioning that the methodology used does not allow the use of CNFs or graphite at high concentrations; it also promotes compounds with polymer-coated particles, so it is complicated to generate the tunneling effect [32,33,34].

### 3.2. Power Generation in MFCs

#### 3.2.1. Power Output

As shown in Figure 7, the maximum power density reached was 35.51 ± 0.02 mW/m^3^, 232.32 ± 0.02 mW/m^3^ and 90.78 ± 0.03 mW/m^3^ corresponding to the synthesized composites based on PU, PU /Graphite 0.5% and PU/CNF 1%, respectively. The values obtained represent an increase of 6.5 (PU/Graphite 0.5%) and 2.5 times (PU/CNF 1%) the power density achieved by polyurethane foam (PU) in its pure state (this material could be generating an electric current due to the exposure of the stainless steel collector to the anodic solution and the microorganisms in the system). These results are similar to those reported by Liu et al. [35] (486.6 mW/m^3^) but higher than those obtained by Chaijak et al. [36] (3.26 µW/m^3^). This confirms the effect that embedded materials (graphite and CNFs) have on the overall performance of MFCs, by increasing the electrical conductivity and surface area of the synthesized anodes. Finally, it is important to highlight the behavior shown in this figure, where a constant increase in power density is observed as a function of the oxidation of the substrate by microbial action; as the concentration of the substrate in the system decreases, the power density generated in the MFCs also decreases [37].

#### 3.2.2. Coulombic Efficiency

In bioelectrochemical systems, the coulombic efficiency determines the fraction of electrons recovered as electrical energy from the oxidation of a substrate. The coulombic efficiency obtained by the composites synthesized based on PU, PU/Graphite 0.5% and PU/CNF 1% was 1.73, 5.87 and 4.41%, respectively (Table 3). As mentioned by Zhang et al. [38], the use of substrates rich in organic matter of high complexity in MFCs can severely diversify the number of metabolic pathways (methanogenesis, nitrification, denitrification, sulfate-reduction, among others) carried out by microorganisms in the system. The latter strongly affects the overall performance of MFCs, such as the municipal wastewater used in the present study [39,40]. Despite the above, the results demonstrate the feasibility of using this type of material as anodes in a bioelectrochemical system to generate electric power.

### 3.3. COD Removal in MFCs

Figure 8 shows the COD removal efficiency in dual-compartment MFCs. It can be observed that the PU, PU/Graphite 0.5% and PU/CNF 1% composites achieved a removal efficiency of 86.04, 51.38 and 68.62% of the COD present in the wastewater, respectively. The morphological analysis of the synthesized materials indicates a decrease in the diameter of the pores of the polyurethane matrix when embedding the graphite and CNF particles, causing an increase in the surface area of the manufactured composites but decreasing the transfer rate of the substrate to the microorganisms that inhabit the support. Various authors identify the distribution of the substrate and its interaction with microorganisms in the system as one of the main variables that decreases the generation of electrical energy and the removal of organic matter in MFCs [41,42]. This phenomenon could explain the results obtained, which are inconclusive because such behavior can be modified when using systems with assisted stirring or continuous feeding because, in this study, the experiments were carried out in a batch regime [43].

### 3.4. Polarization and Power Curves

Figure 9 shows the electrochemical characterization of the MFC that produced a better performance (PU/Graphite 0.5%) during the experiments. The maximum power density reached was 302.99 mW/m^3^ at a 550.45 mA/m^3^ current density. These values indicate the maximum possible performance of the MFC once the value of the external resistance is equaled to the value of the internal resistance (1051.11 Ω, value calculated from the slope of the potential curve) of the system, maintaining the same configuration, substrate and materials used for its construction and operation [44,45].

## 4. Conclusions

The results obtained in this study, such as the maximum power density (232.32 mW/m^3^ and 90.78 mW/m^3^), coulombic efficiency (5.87 and 4.41%), COD removal efficiency (51.38 and 68.62%) and internal resistance (1051.11 Ω, value calculated only for the first mentioned synthetized material) of the MFCs packed with the PU/Graphite 0.5% and PU/CNF 1% composites, demonstrate the feasibility of using polyurethane, graphite and carbon nanofiber composites in dual-compartment microbial fuel cells for electric power generation and contaminated effluent treatment. The low production cost of this type of material (considering the low graphite and CNF loading percentages in the polyurethane matrix) makes its use and potential scaling even more attractive, so this will be our research group’s main objective of future study. Finally, embedding other more conductive materials or functionalizing the materials already in use could considerably increase the composites’ electrical conductivity, decreasing the system’s internal resistance.

## Figures and Tables

**Figure 1 polymers-15-04177-f001:**
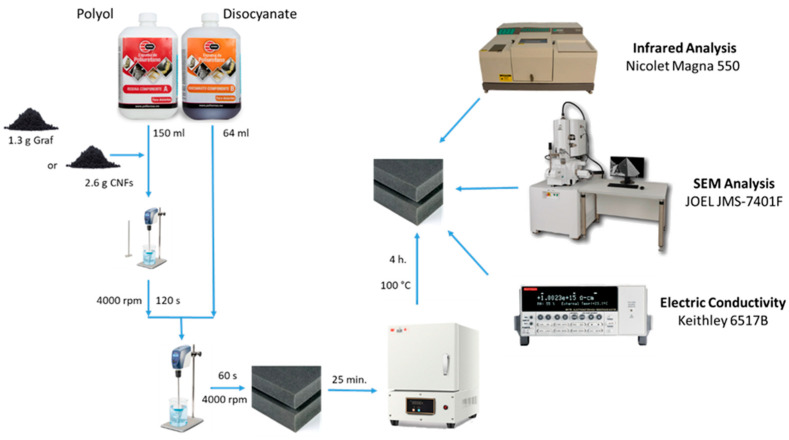
Schematic diagram of the manufacturing process of foamed materials used as anodes.

**Figure 2 polymers-15-04177-f002:**
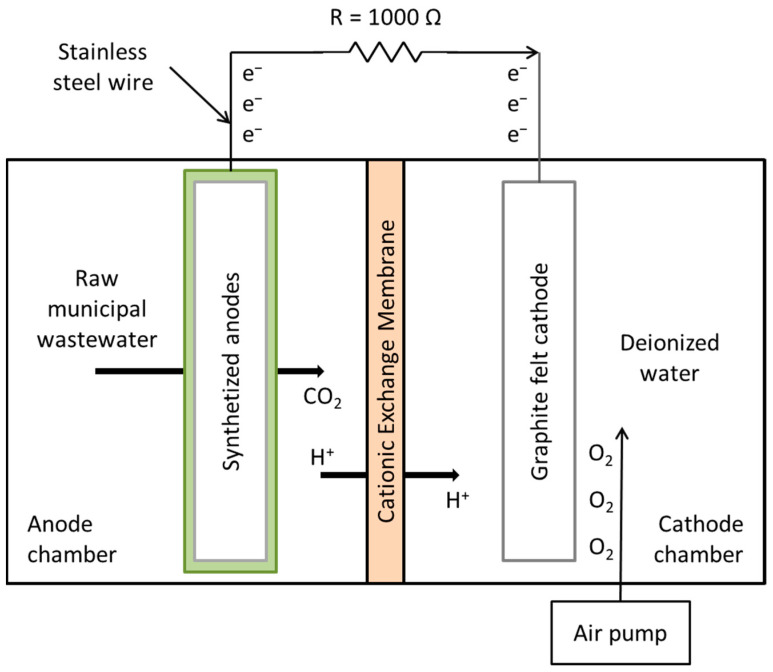
Schematic illustration of the MFCs used in the present study.

**Figure 3 polymers-15-04177-f003:**
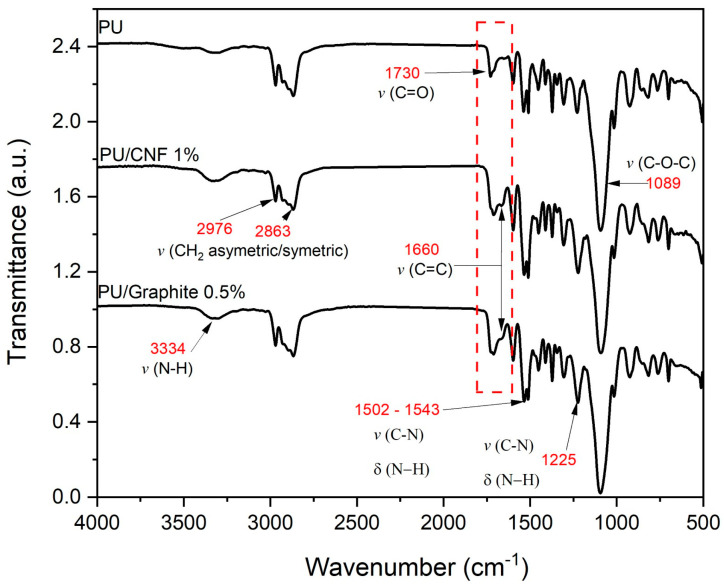
FTIR spectrograms of PU, PU/Graphite 0.5% wt./wt. and PU/CNF 1% wt./wt.

**Figure 4 polymers-15-04177-f004:**
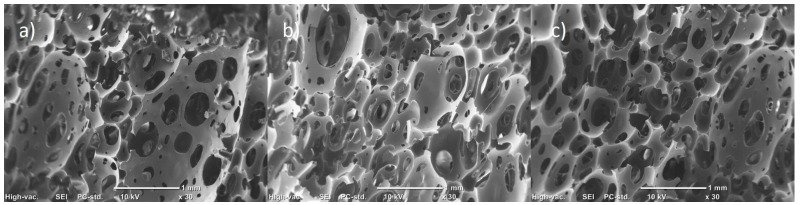
SEM micrographs of pristine PU at ×30 magnification: (**a**) zone 1; (**b**) zone 2; (**c**) zone 3.

**Figure 5 polymers-15-04177-f005:**
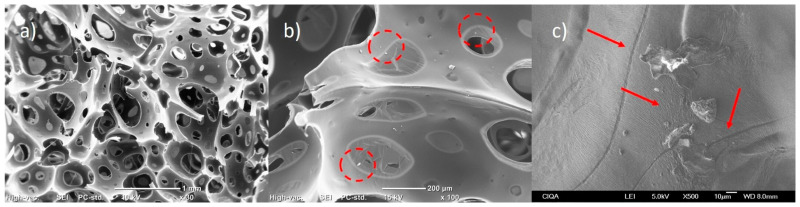
SEM micrographs of PU/Graphite 0.5% wt./wt. composites at different magnifications: (**a**) ×30; (**b**) ×100; (**c**) ×500.

**Figure 6 polymers-15-04177-f006:**
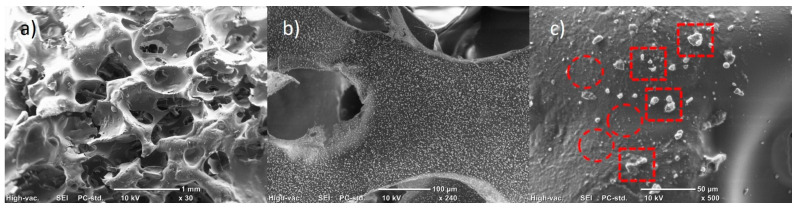
SEM micrographs of PU/CNF 1% wt./wt. composites at different magnifications: (**a**) ×30; (**b**) ×100; (**c**) ×500.

**Figure 7 polymers-15-04177-f007:**
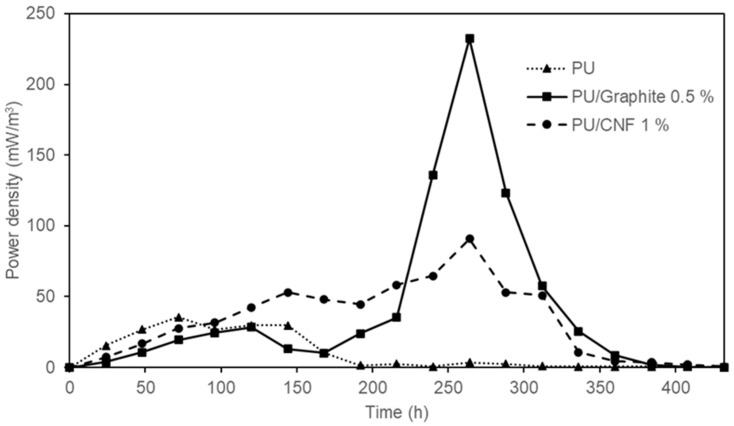
Power density generated by the MFC devices with PU, PU/Graphite 0.5% and PU/CNF 1% composites as anodes.

**Figure 8 polymers-15-04177-f008:**
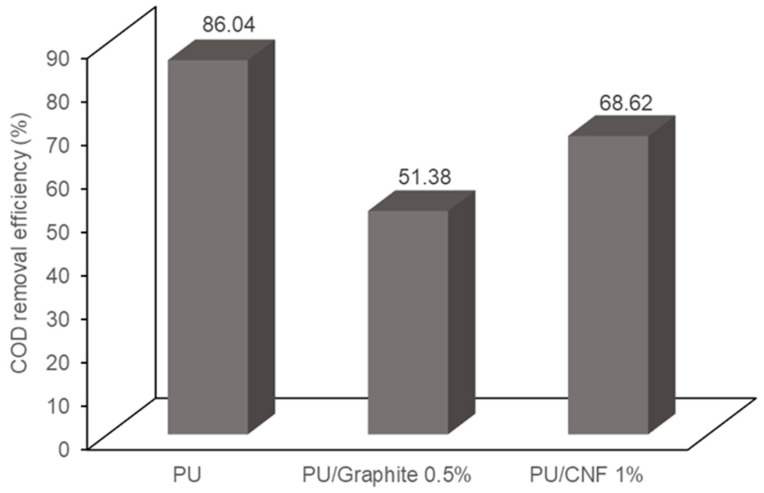
COD removal efficiency in the MFCs used in the present study.

**Figure 9 polymers-15-04177-f009:**
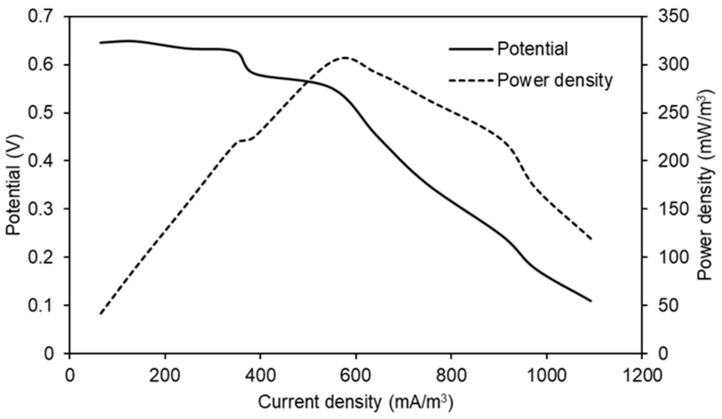
Potential and power curves as a function of current density obtained in the MFC that used PU/Graphite 0.5% composite anode.

**Table 1 polymers-15-04177-t001:** Amount of reactants used in anode preparation.

Anode	Graphite, g	CNF, g	Polyol, mL	Diisocyanate, mL
PU (blank)	-	-	150	64
PU/Graphite 0.5%	1.3	-	150	64
PU/CNF 1%	-	2.6	150	64

**Table 2 polymers-15-04177-t002:** Physicochemical analysis of raw municipal wastewater.

Parameters	Raw Municipal Wastewater
Color	Dark grey
Odor	Strong pungent
pH	9.24
Electrical conductivity	1.12 mS/cm
Chemical oxygen demand	703.13 mg/L
Temperature	24–27 °C

**Table 3 polymers-15-04177-t003:** Coulombic efficiency obtained by the MFC devices.

Anode	Coulombic Efficiency (%)
PU (blank)	1.73
PU/Graphite 0.5%	5.87
PU/CNF 1%	4.41

## Data Availability

The data that support the findings of this study are available from the corresponding author upon request.

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
