# Peer review of "Embedded Graphite and Carbon Nanofibers in a Polyurethane Matrix Used as Anodes in Microbial Fuel Cells for Wastewater Treatment"

_polymers, 2023, doi:10.3390/polym15204177_

Round 1
Reviewer 1 Report
We received a paper from Polymers (polymers-2619344) with the title “Embedded graphite and carbon nanofibers in a polyurethane matrix used as anodes in microbial fuel cells for wastewater treatment”. After carefully checking the manuscript, several aspects need to be revised before it goes to be published.
11. In the abstract, if the authors state the abbreviation, it should be added with the full term of it. There are abbreviations that are not followed with the full term i.e. COD and CCM. Please revise it.
22. Since the composites and MFC are the main topics in the present study, add these terms in the keywords.
33. In the last introduction, please add the gap between the previous work and why the present study can fulfill it.
44. In the introduction, it is stated a few narrations about the importance of coating technology in various aspects. Add the present references in the study to improve the introduction strength to the readers.
55. Add the figure that shows a schematic study of the raw material used, methods, and characterization process on it.
66. In Fig. 2, the FTIR graph was provided with 2 specimens. However, the graph should be added with a control sample so that the reader can understand what the differences between filler addition and without of it.
77. In Fig. 3, it should be added with narration in the figure caption, what (a) , (b) and (c) mean?
88. Same situation with Fig 4 and Fig. 5
99. In Fig.6, it is shown that in time over 250 (h), the significant jumping on the graph occurred especially for PU/Graphite 0.5%. why this occur?
110. The paper should add the basic calculation related to coulumbic efficiency (%) and how to generate that value.
111. In Fig.7 it is shown that the control sample (PU) has the highest value compared with other specimens. Why this occurred? It is true that the efficiency of the non-pure sample is lower due to the filler effect?
112. In the conclusion, please add all the important values of the testing and the results. Add the future applications of the best specimen based on the results.
minor spelling and over all fine.
Author Response
REFEREE 1:
Comment 1: In the abstract, if the authors state the abbreviation, it should be added with the full term of it. There are abbreviations that are not followed with the full term i.e., COD and CCM. Please revise it.
Response: We have added the corresponding terms in the mentioned section, as suggested by the reviewer
Comment 2: Since the composites and MFC are the main topics in the present study, add these terms in the keywords.
Response: We have added the mentioned terms as keywords as suggested by the reviewer.
Comment 3: In the last introduction, please add the gap between the previous work and why the present study can fulfill it.
Response: In the last paragraph of the introduction (lines 88-92), we added the information the reviewer requested, thanking him for his comment.
Comment 4: In the introduction, it is stated a few narrations about the importance of coating technology in various aspects. Add the present references in the study to improve the introduction strength to the readers.
Response: We have added two references that highlight the importance of coatings in the development of conductive materials for application in bioelectrochemical systems, and we have revised the rest of the document to ensure that the rest of the references are up to date as suggested by the reviewer.
Comment 5: Add the figure that shows a schematic study of the raw material used, methods, and characterization process on it.
Response: We have added the schematic requested by the reviewer to the document.
Comment 6: In Fig. 2, the FTIR graph was provided with 2 specimens. However, the graph should be added with a control sample so that the reader can understand what the differences between filler addition and without of it.
Response: We have added the spectrogram of the polyurethane foam in its pure state to Figure 3 (previously Figure 2), as requested by the reviewer.
Comment 7: In Fig. 3, it should be added with narration in the figure caption, what (a), (b) and (c) mean?
Response: We have added the description as suggested by the reviewer.
Comment 8: Same situation with Fig 4 and Fig. 5.
Response: We have added the descriptions as suggested by the reviewer.
Comment 9: In Fig.6, it is shown that in time over 250 (h), the significant jumping on the graph occurred especially for PU/Graphite 0.5%. why this occur?
Response: This type of behavior is common in bioelectrochemical systems in batch mode, whose substrate has a low concentration of organic matter (< 1 g COD/L). In the graph you can see a constant increase in the power density generated, as long as there is enough substrate to be consumed by the microorganisms in the system. As this substrate begins to be exhausted, the power density decreases in the same way. The above was added to the document and correctly referenced with the objective of strengthening the work.
Comment 10: The paper should add the basic calculation related to coulombic efficiency (%) and how to generate that value.
Response: Equation 2 and lines 169-173 contain the information requested by the reviewer. Likewise, we improved the writing to facilitate understanding the calculations carried out.
Comment 11: In Fig.7 it is shown that the control sample (PU) has the highest value compared with other specimens. Why this occurred? It is true that the efficiency of the non-pure sample is lower due to the filler effect?
Response: We consider that the pore size of the synthesized materials plays an important role in the transfer rate of the substrate to the microorganisms that inhabit the supports. Despite this, we understand the reviewer's concern and modified this statement to avoid controversial comments.
Comment 12: In the conclusion, please add all the important values of the testing and the results. Add the future applications of the best specimen based on the results.
Response: We have added the results and described the future continuation of this work in the conclusions section as requested by the reviewer.
Reviewer 2 Report
Please refer to the attachment for the detailed comments.

A few typos and wording issues were observed. However, the number of these minor issues is very small. I think the manuscript is well-written and clear
Author Response
REFEREE 2:
Comment 1: It is not clear how the additives (graphite and CNF) affected the electrical conductivity of the PU based electrodes. Could the authors provide this data for further insight into the material properties?
Response: We have added a new section in the results section (point 3.1.3) where the effect of incorporating the materials of interest (graphite and CNF) in the polyurethane matrix is highlighted. The above was described to strengthen the document as suggested by the reviewer.
Comment 2: The manuscript does not provide information regarding the testing of different weight percentages of graphite and CNF. Could the authors elaborate on the rationale behind choosing 0.5 wt% and 1 wt%, respectively?
Response: Preliminary tests allowed us to establish the maximum percentage of loading of the conductive materials in the polyurethane matrix, observing that at concentrations higher than those used in this study, the synthesized material considerably increased its density, producing instability and the collapse of the manufactured composite. The above was added to the document to enrich the work.
Comment 3: The manuscript suggests power generation from the PU electrode. Considering the non-conductive nature of PU, could the authors clarify this observation? Is there a possibility that the stainless-steel collector might have been exposed to anodic electrolytes, contributing to the current observed?
Response: Since there is no isolation between the electron collector and the anodic solution, we consider the reviewer's comment correct, describing this phenomenon in the manuscript and we thank the reviewer for his feedback.
Comment 4: The time point for the polarization and power curve measurement in Figure 8 is not specified. Were these curves measured around the 260-hour mark when the MFC exhibited peak power output?
Response: The polarization and power curves were determined after feeding the system (CCM) with a new sample of raw municipal wastewater, waiting for the stabilization of the open circuit voltage (VCA) prior to the application of external resistances. The above was added to the document with the aim of enriching the work.
Comment 5: There appears to be an error in the time mentioned in the header and footer, which reads “2021”.
Response: We could not find this error, observing, in our case, 2023 as the year of publication. If this error continues, we trust that it can be resolved during the editing process, as mentioned by the reviewer.
Comment 6: Page 1, abstract. The abbreviation “CCM” is used but not expanded. Could the authors clarify what “CCM” stands for?
Response: The abbreviation “CCM” is a translation error, so we thank the reviewer for underlining the error.
Comment 7: Page 3, section 2.4, line 4. Typo of “catolytic”.
Response: We replace the word “catolytic” with the term cathodic.
Comment 8: Page 6, last paragraph, line 6, and page 8, section 3.3, line 11. The words “said” could be more appropriately replaced with “such”.
Response: We make the correction as mentioned, thanking the reviewer for the comment.
Comment 9: Page 8, first paragraph, lines 3-4. The phrase “to use this type of support as final electron acceptors” may require revision for clarity. It is presumed that the authors are referring to the composite electrodes as electron acceptors for the bacterial cells.
Response: We have removed the indicated phrase from the final text, with the aim of generating clarity in the document and avoiding controversial statements as suggested by the reviewer, greatly appreciating the feedback.
Round 2
Reviewer 1 Report
After carefully check the manuscript, I recommend the present version to be accepted.